# MCAK Inhibitors Induce Aneuploidy in Triple-Negative Breast Cancer Models

**DOI:** 10.3390/cancers15133309

**Published:** 2023-06-23

**Authors:** John C. Smith, Stefan Husted, Jay Pilrose, Stephanie C. Ems-McClung, Jane R. Stout, Richard L. Carpenter, Claire E. Walczak

**Affiliations:** 1Medical Sciences, Indiana School of Medicine—Bloomington, Bloomington, IN 47405, USA; jcs12@iu.edu (J.C.S.); scems@indiana.edu (S.C.E.-M.); janstout@indiana.edu (J.R.S.); richcarp@indiana.edu (R.L.C.); 2LabCorp Drug Development Indianapolis, Indianapolis, IN 46214, USA; sthusted03@gmail.com; 3Catalent Pharma Solutions Bloomington, Bloomington, IN 47403, USA; jay.pilrose@gmail.com

**Keywords:** triple-negative breast cancer, MCAK/KIF2C, aneuploidy, chromosomal instability, drug discovery

## Abstract

**Simple Summary:**

Triple-negative breast cancer (TNBC) is the most lethal breast cancer subtype with few treatment options available. Standard of care for TNBC involves the use of taxanes, which are initially effective, but dose limiting toxicities are common and patients often relapse with resistant tumors. Specific drugs that produce taxane-like effects may be able to improve patient quality of life and prognosis. In this study we identify three novel inhibitors of the Kinesin-13 MCAK. MCAK inhibition induces aneuploidy, similar to cells treated with taxanes. We demonstrate that MCAK is upregulated in TNBC and is associated with poorer prognoses. These MCAK inhibitors reduce the clonogenic survival of TNBC cells, and the most potent of the three inhibitors, C4, sensitizes TNBC cells to taxanes, similar to the effects of MCAK knockdown. This work will expand the field of precision medicine to include aneuploidy-inducing drugs that have the potential to improve patient outcomes.

**Abstract:**

Standard of care for triple-negative breast cancer (TNBC) involves the use of microtubule poisons such as paclitaxel, which are proposed to work by inducing lethal levels of aneuploidy in tumor cells. While these drugs are initially effective in treating cancer, dose-limiting peripheral neuropathies are common. Unfortunately, patients often relapse with drug-resistant tumors. Identifying agents against targets that limit aneuploidy may be a valuable approach for therapeutic development. One potential target is the microtubule depolymerizing kinesin, MCAK, which limits aneuploidy by regulating microtubule dynamics during mitosis. Using publicly available datasets, we found that MCAK is upregulated in triple-negative breast cancer and is associated with poorer prognoses. Knockdown of MCAK in tumor-derived cell lines caused a two- to five-fold reduction in the IC_50_ for paclitaxel, without affecting normal cells. Using FRET and image-based assays, we screened compounds from the ChemBridge 50 k library and discovered three putative MCAK inhibitors. These compounds reproduced the aneuploidy-inducing phenotype of MCAK loss, reduced clonogenic survival of TNBC cells regardless of taxane-resistance, and the most potent of the three, C4, sensitized TNBC cells to paclitaxel. Collectively, our work shows promise that MCAK may serve as both a biomarker of prognosis and as a therapeutic target.

## 1. Introduction

Most tumor cells have highly heterogeneous chromosome content with changes in both copy number and structural abnormalities on individual chromosomes [1]. A large subset of tumor cells exhibits chromosomal instability (CIN), causing cells to gain and lose chromosomes with each division. CIN ultimately leads to aneuploidy in which the resulting daughter cells have an irregular number of chromosomes [2]. CIN can also induce the formation of micronuclei, which can be especially damaging, as micronuclei can undergo dramatic DNA rearrangements through chromothrypsis [3] and may result in epigenetic changes in daughter cell nuclei [4,5]. While CIN and aneuploidy are generally detrimental to cells, CIN can benefit tumor cells, as it can lead to tumor evolution by retention of favorable karyotypes [6,7].

Despite the negative consequences of CIN and aneuploidy, emerging work suggests that CIN offers therapeutic opportunities in the treatment of cancer [8]. An analysis of gastric cancer samples treated with neoadjuvant chemotherapy revealed that CIN levels above a certain threshold conferred better responses to neoadjuvant chemotherapy [9]. A more thorough understanding of the mechanisms of CIN may also reveal better therapeutic options for patients. Recent work has identified distinct molecular subtypes of CIN that are significantly correlated with vulnerability to existing drugs [10,11]. Subtypes of chromosomal instability have also been shown to predict sensitivity to novel targets [10]. High levels of CIN are also associated with reduced tumor fitness and better patient prognoses [9,12].

An emerging idea is that inducing lethal levels of aneuploidy through modulation of CIN may be a valid mechanism to kill cancer cells [1]. The anti-microtubule (MT) taxanes are powerful chemotherapeutics used in breast, ovarian, and other cancers [13]. Early work suggested that taxanes induce mitotic arrest, causing cells to eventually undergo apoptosis [14]. However, in tumors treated with taxanes, there is no significant increase in cells in the metaphase of mitosis, making this hypothesis unlikely [15,16]. More recent studies show that taxane efficacy requires cells to transit through mitosis and clinical taxane efficacy may be due to an induction of lethal aneuploidy [17]. Unfortunately, taxanes carry numerous harmful side effects, including neuropathy [18,19], and cancers often develop resistance [20]. Therefore, developing targeted inhibitors that produce taxane-like effects without damaging neurons could be a valuable clinical approach.

The Kinesin-13 MCAK (mitotic centromere-associated kinesin)/Kif2C, is a powerful MT destabilizing enzyme [21,22] whose inhibition leads to an increase in MT stability in interphase cells [23,24,25] and in chromosome missegregation/aneuploidy during mitosis [24,26,27,28]. Altering MCAK levels in cells has been used to modulate CIN [29,30]. Expression of a dominant-negative MCAK led to increased CIN, which correlated with an increase in metastasis in xenograft models [31]. We and others have also shown that modulating MCAK levels inhibit cell migration through dysregulating cell polarity and focal adhesion turnover [32,33,34]. These data suggest that MCAK levels may need to be tightly modulated to maintain proper cellular homeostasis. MCAK expression is upregulated in multiple cancers [35,36], and it is one of the cell cycle regulated genes highly expressed across multiple cancer types [37]. MCAK is also part of the PAM50 gene signature for breast cancer [38]. High MCAK levels correlate with poor prognosis in multiple cancers and are associated with increased invasion and metastasis [37,39,40,41,42,43]. MCAK has also been shown to mediate resistance to paclitaxel [44,45]. One idea is that tumors become dependent on high MCAK levels to limit aneuploidy; thus, targeting MCAK in these cells may lead to excessive chromosome missegregation and aneuploidy. MCAK inhibition is similar to taxane treatment with some distinct differences in mitosis [46], and MCAK inhibition is synergistic with anti-MT agents [47]. MCAK overexpression has been linked to paclitaxel resistance, and loss of MCAK has been shown to reverse tubulin mutation-induced paclitaxel resistance [44,45]. MCAK is not highly expressed in neurons, suggesting that it could be an effective target without inducing the neurological effects associated with MT-targeting drugs such as taxanes.

In this study, we assessed the potential of targeting MCAK in cell-line models of triple-negative breast cancer (TNBC) because MCAK/Kif2C is highly expressed in breast cancer [48,49,50], and because anti-MT agents are often first-line therapies for TNBC, which frequently become resistant to these drugs [51]. We found that high MCAK expression correlates with poor prognosis and increased metastasis. Knockdown of MCAK is synergistic with taxanes for growth inhibition in breast tumor cells and induces micronuclei formation in taxane-resistant cells. We developed screening assays and identified multiple MCAK inhibitors that reduce clonogenic growth and induce aneuploidy in taxane-resistant cells, and the most potent inhibitor sensitizes cells to taxanes in cell growth assays. Together these studies support the idea that MCAK may be a valuable target for therapeutic development.

## 2. Materials and Methods

### 2.1. Bioinformatics Analysis

Publicly available data from The Cancer Genome Atlas (TCGA) was used to generate a KIF2C expression dataset in normal (N = 112) and breast cancer (N = 1100) samples. Samples were stratified into Normal or Cancer, according to receptor status, or according to PAM50 subtype and plotted in GraphPad Prism (GraphPad Software Inc., Boston, MA, USA). For overall survival, the TCGA breast cancer samples (N = 1100) were stratified into high and low KIF2C by median split and used to generate a Kaplan–Meier plot in GraphPad Prism. KIF2C expression data (N = 1121) from the GSE47561 database was accessed and stratified by median split to generate a Kaplan–Meier curve for distant metastasis-free survival. Data from the GSE25066 database were used to generate a dataset of KIF2C expression vs. pathological complete response (pCR; N = 196) and residual disease (RD; N = 312).

### 2.2. Cell Lines and Culture

MDA-MB-231 (Cellosaurus CVCL_0062), PTXR231 (paclitaxel-resistant, Cellosaurus CVCL_4Z64) [52], RPE-1 (Cellosaurus CVCL_4388), HeLa (Cellosaurus CVCL_0030), and U2OS (CVCL_0042) cells were maintained in DMEM (Invitrogen) containing 10% FBS and 50 mg/mL penicillin/streptomycin (DMEM media). MCF10A (Cellosaurus CVCL_0598) cells were maintained in DMEM/F12 (Invitrogen, Waltham, MA, USA) containing 10% FBS, 50 mg/mL penicillin/streptomycin, 10 µg/mL insulin, 0.5X sodium pyruvate, 20 ng/mL EGF, and 0.5 µg/mL hydrocortisone (DMEM/F12 media). E8.1 HeLa cells were maintained in DMEM with 10% tetracycline-free FBS as previously described [53,54]. To maintain cell-line integrity, cultures were not used for more than 20 passages. Identity of all cell lines was confirmed by STR profiling (LabCorp, Burlington, NC, USA). The relative expression level of MCAK in different cell lines was compiled from the Human Atlas project, the GSE10890 database and our own qPCR data (Appendix A). 

### 2.3. MTT Assays

Cells were seeded in 96-well plates at 2000–10,000 cells per well and allowed to adhere for 24 h. For knockdown experiments, cells were transfected with 10 nM of the indicated siRNA (Neg2: UGGUUUACAUGUUGUGUGA, MCAK siRNA1: ACCAUUACUGCGUUGGAUC [47], MCAK siRNA2: GAGAGCUCAGGAGUAUGACAGUAGU) with Lipofectamine RNAiMax (ThermoFisher, Waltham, MA, USA) according to the manufacturer’s instructions in a total volume of 100 µL using Opti-MEM media (Invitrogen). After 24 h, the siRNA media was removed and cells were treated with the indicated concentrations of drugs for 48 h. Following drug treatment, the drug media was aspirated and replaced with 50 µL of 0.5 mg/mL 3-(4,5-dimethylthiazol-2-yl)-2,5-diphenyltetrazolium bromide (MTT) in media. Following a 3-hr incubation, MTT media was aspirated and replaced with 150 µL of DMSO and placed on a shaker for 10 min to dissolve the formazan crystals. Plates were then read on the Synergy H1 microplate reader (BioTek Instruments, Winooski, VT, USA) at 570 and 650 nm. The 650 nm reading was subtracted from the 570 nm reading to generate delta values. Duplicate conditions were averaged and then normalized against control samples of cells treated with an equivalent concentration of DMSO. All IC_50_ values were calculated using the log (inhibitor) vs. response (three parameters) analysis option in GraphPad Prism (GraphPad Software Inc.). 

### 2.4. Assays to Measure Aneuploidy

MCAK knockdown was carried out as described above with volumes scaled for a 6-well plate format. Cells were plated 40,000 cells/mL on poly-L-lysine coated coverslips in 6-well plates and allowed to adhere for 24 h. Cells were transfected with 10 nM Neg2 siRNA, MCAK siRNA1, or MCAK siRNA2 in Opti-MEM using the Lipofectamine™ RNAiMAX Transfection Reagent according to the manufacturer’s instructions. After 24 h, the transfection media was removed and cells were treated with 3 µM Cytochalasin B (Sigma-Aldrich, St. Louis, MO, USA) in DMEM media for 24 h in which any cells that progressed through mitosis would undergo cytokinesis failure, resulting in the formation of binucleate cells. Cells were then fixed and processed for immunofluorescence as outlined below. The percentage of cells with micronuclei were quantified by counting 100 binucleate cells for each treatment and categorizing them as micronucleated or non-micronucleated in at least three independent experiments. For the inhibitor-induced aneuploidy assays, cells were plated as described above for 6 well plates and allowed to recover for 48 h. Cells were treated with 50 µM inhibitor and 3 µM Cytochalasin B for 24 h. Cells were then fixed and processed as described below. For each experiment, 100 binucleate cells were counted and classified as either normal or containing one or more micronuclei. For assessment of lagging chromosomes in the inducible MCAK knockout line, cells were plated in T25 flasks and left untreated or treated with 1 µg/mL doxycycline hyclate for five days. Cells were then plated at 40,000 cells/mL on coverslips in 6-well plates and allowed to recover for 48 h prior to treatment with 50 µM of inhibitor for 24 h. Cells were then fixed and processed as described below. For each experiment, 100 anaphase cells were counted per condition and classified as normal or containing one or more lagging chromosomes.

### 2.5. Immunofluorescence Staining and Imaging 

For immunofluorescence assays, cells on coverslips were pre-extracted with a solution of 0.5% Triton™ X-100 in MT Stabilizing Buffer (MTSB: 4M Glycerol, 100 mM PIPES pH 6.8, 1 mM EGTA [ethylene glycol-bis(2-aminoethylether)-N,N,N′,N′-tetraacetic acid], 5 mM MgCl2) for two min and then fixed in a solution of 4% formaldehyde in PHEM (60 mM Pipes, 25 mM HEPES, 5 mM EGTA, 1 mM MgCl2, pH 6.9) for 20 min. Coverslips were blocked in Ab-Dil (20 mM Tris pH 7.5, 150 mM NaCl, 2% bovine serum albumin (BSA), 0.1% Triton™ X-100, 0.1% NaN3) for 30 min. Cells were incubated in their indicated primary antibodies for 30 min, washed three times with TBS-Tx (20 mM Tris pH 7.5, 150 mM NaCl, 0.1% Triton X-100) and then incubated in their indicated secondary antibodies for 30 min. Cells were washed again three times with TBS-Tx and incubated with 10 µg/mL Hoechst 33342, washed three times, and mounted on slides with ProLong™ Diamond Antifade Mountant (ThermoFisher). The primary antibodies used were mouse DM1α anti-alpha-tubulin (1/1000; Sigma T9026) and a rabbit anti-human MCAK (hMCAK, 1 µg/mL) raised against amino acids 2–154 as previously described [47]. The secondary antibodies used were Alexa Fluor™ 488 donkey anti-rabbit (1 µg/mL; Invitrogen A-21206) and Alexa Fluor™ 594 donkey anti-mouse (1 µg/mL; Invitrogen A-21203). Images were acquired on a Nikon 90i microscope equipped with a Plan Apo VC 60X/1.40 NA oil objective and controlled by MetaMorph^®^ (Molecular Devices, San Jose, CA, USA). Images for all cell-based assays were collected using the Multi-Dimensional Acquisition feature on the MetaMorph^®^ version 7.8.2.0 software for Z-stacking and multi-channel imaging. Conditions within experiments were captured with equivalent exposures and numbers of Z-slices. Images were opened in FIJI Version 1.53c (FIJI Is Just ImageJ, NIH) [55] and Z-projected using the max project option. All images were normalized to equivalent minimum and maximum gray values.

### 2.6. Western Blotting

Cells were seeded in 6-well plates at 40,000–80,000 cells per well and allowed to adhere for 24 h. Cells were transfected with 10 nM of the indicated siRNA for 24 h and allowed to outgrow for an additional 48 h prior to lysate collection. Cells were trypsinized, washed twice with phosphate-buffered saline (PBS, 137 mM NaCl, 2.7 mM KCl, 10 mM Na_2_HPO_4_, 1.8 mM KH_2_PO_4_, pH 7.2), counted and pelleted at 400× *g* for 10 min before resuspending at 10,000 cells per µL in RIPA lysis buffer (150 mM NaCl, 50 mM Tris, 1% Nonidet P-40, 0.5% Sodium deoxycholate, 0.1% SDS) and rotating at 4 °C for 1 h. The lysate was then spun at 12,000 RPM for 15 min at 4 °C. Lysates were transferred to a new tube, diluted 1:1 with 2X sample buffer (0.125 M Tris pH 6.8, 4% SDS, 20% glycerol, 4% ß-mercaptoethanol, and a trace amount of bromophenol blue) and boiled for 5 min at 95 °C. Equal amounts of cell lysates were electrophoresed on 10% (*v*/*v*) SDS–PAGE gels and transferred to nitrocellulose (Schleicher & Schuell). Blots were incubated in AbDil-T (20 mM Tris pH 7.5, 150 mM NaCl, 0.1% Tween-20, 2% BSA, and 0.1% sodium azide) and probed with mouse DM1α (1/5000) and rabbit anti–hMCAK-154 (1 μg/mL) [47] diluted in Abdil-T. Secondary antibodies were used at 1 μg/mL for goat anti-rabbit and sheep anti-mouse linked horseradish peroxidase (Invitrogen). Blots were developed with SuperSignal West Pico Chemiluminescent Substrate (Pierce).

### 2.7. Recombinant Protein Expression and Purification

The pFB6HmCit vector was generated by PCR amplification of mCitrine (mCit) from pRSETA-5′mCit [56] and subcloning the fragment into the BamHI/SacI sites of pFB6H [57]. The pFB-6HFRET2-DEST vector was created by subcloning DNA encoding the DEST cassette and mCerulean (mCer) from the pFBFRET2-DEST vector [56] into the SacI/HindIII sites of pFB6HmCit such that mCit was 5′ and mCer was 3′. The pENTR-hMCAK construct was generated by amplifying the coding sequence of human MCAK by PCR and inserting it into a pENTR entry clone using the pENTR^TM^/D-TOPO kit (ThermoFisher). The pFB-6HF2hMCAK construct was created from pFB-6HFRET2-DEST vector and pENTR-hMCAK constructs using Gateway^®^ cloning technology (Invitrogen). All constructs were verified by restriction digest and by sequencing. Baculovirus stocks were generated from pFB-6HF2hMCAK constructs using the Bac-to-Bac^®^ System (Invitrogen).

FRET-tagged hMCAK protein (FMCAK) was expressed and purified from baculovirus infected High Five insect cells using the Bac-to-Bac expression system. High Five cells were cultured at 27 °C in Express Five SFM growth media supplemented with 2 mM L-glutamine and 100 µg mL-1 anti-microbial mixture. High Five cells (400 mL) expressing FMCAK were pelleted, frozen in liquid nitrogen, and stored at −80 °C until use. Frozen cell pellets were resuspended by pipetting in 40 mL of ice-cold lysis buffer (BRB80 (80 mM K-Pipes pH 6.8, 1 mM EGTA, 1 mM MgCl_2_), 120 mM KCl, 2 mM ATP, 1 mM DTT, 0.5% Triton-X100, 1 μg/mL leupeptin/pepstatin/chymostatin (LPC), 1 mM PMSF) [21]. The lysate was sonicated on ice for 20 s at 20% output on a Branson sonifier, clarified by centrifugation at 20,000 rpm (48,385× *g*) for 30 min in a JA25.50 rotor at 4 °C (Beckman AvantiTM J-25, Indianapolis, IN, USA), filtered through a 0.8/0.2 AcroPak Filter Unit (Pall, Port Washington, NY, USA), and then loaded onto a HiTrap SP Sepharose column (Cytiva Life Sciences, Marlborough, MA, USA). The column was washed with 5 column volumes of 12% salt FPLC buffer (BRB20 (20 mM K-Pipes pH 6.8, 1 mM MgCl_2_, 1mM EGTA), 120 mM KCl, 0.1 mM EDTA, 1 mM DTT, 10 µM Mg-ATP, 0.1 µg/mL LPC), and the proteins were eluted from the column with a 120 to 1000 mM KCl gradient in FPLC buffer. The peak fractions of FMCAK (~350 mM KCl), as visualized by SDS-PAGE, were pooled, filtered through a 0.2 µm filter unit (Millex), and loaded on a Superose 6 gel filtration column (Cytiva Life Sciences), which was equilibrated in 30% salt FPLC buffer (BRB20, 300 mM KCl, 0.1 mM EDTA, 1 mM DTT, 10 µM Mg-ATP, 0.1 µg/mL LPC). Peak fractions from the Superose 6 column were pooled, supplemented with solid sucrose to 10% (*w*/*v*), aliquoted, frozen in liquid nitrogen, and stored at −80 °C. Protein concentration was determined by gel densitometry using BSA as a standard. All concentrations of FMCAK are reported as its dimer molar amount.

FRET control protein [56] (FCP) and Aurora B/INCENP [56] were expressed as 6His-tagged proteins in BL21 (DE3) bacterial cells, induced with 0.1 mM IPTG for 24 h at 37 °C. Frozen cell pellets were lysed in 20 mL lysis buffer (50 mM phosphate pH 8.0, 300 mM NaCl, 0.1% Tween-20, 10 mM imidazole, 1 mM PMSF, 1 mM benzamidine), 0.5 mg/mL lysozyme, sonicated 3X for 20 s at 20% output, spun down in a Beckman JA 25.5 centrifuge at 18,000 rpm for 20 min, and purified on a Ni-NTA agarose column according to the manufacturer protocol (Qiagen). The column was washed in 10 volumes of column buffer (50 mM phosphate pH 8.0, 500 mM NaCl, 5 mM beta mercaptothanol (βME), 0.1 mM PMSF) and eluted with 10 volumes of column elution buffer (50 mM phosphate pH 7.2, 500 mM NaCl, 0.5 mM βME, 400 mM imidazole) in 1 mL fractions. Aurora B/INCENP was dialyzed into Aurora B buffer (20 mM Tris pH 7.7, 300 mM NaCl, 0.1 mM EDTA, 1 mM DTT), aliquoted, flash frozen in liquid nitrogen, and stored at −80 °C. FCP eluted fractions were pooled and dialyzed into XB buffer (10 mM HEPES pH 7.7, 100 mM KCl, 25 mM NaCl, 50 mM sucrose, 0.1 mM EDTA, 0.1 mM EGTA). Dialyzed FCP was aliquoted, frozen in liquid nitrogen, and stored at −80 °C. Protein concentrations were determined by densitometry using BSA as a standard. Aurora B concentrations are stated as the Aurora B concentration in the Aurora B/INCENP fraction.

### 2.8. Preparation of MT Substrates and Visual MT Depolymerization Assay

Rhodamine tubulin was generated by labeling tubulin with X-rhodamine SE (5 (and-6) carboxy-X-rhodamine succinimidyl ester) dye (ThermoFisher). Labeling stoichiometry was determined using a molar extinction coefficient of 115,000/M cm for tubulin and 78,000/M cm for X-rhodamine [58]. Biotin tubulin was labeled with 2.8 mM Biotin-LC-LC-NHS (Invitrogen), and concentrations were determined using the molar extinction coefficient of 34,000/M cm [58]. MTs were polymerized with 1 mM GTP at 10 µM total tubulin for 40 min at 37 °C from a mixture of unlabeled tubulin, X-rhodamine labeled tubulin, and biotin labeled tubulin at a molar ratio of 10.8 unlabeled: 5 rhodamine labeled: 1 biotin labeled. Polymerized MTs were stabilized by the addition of three sequential additions of paclitaxel at 0.1 µM, 1 µM, and 20 µM, sedimented by spinning at 45,000 rpm (78,246× *g*) for 5 min at 35 °C in a TLA100 rotor in a Beckman OptimaTM MAX-TL ultracentrifuge, and resuspended in BRB49 buffer (49 mM K-PIPES pH 6.8, 1 mM MgCl_2_, 1 mM EGTA) + 1 mM DTT + 20 µM paclitaxel [59]. MT polymer concentrations are reported in tubulin dimer amounts and are calculated by absorbance at 280 nm using the molar extinction coefficient of 115,000/M cm for tubulin.

Visual MT depolymerization assays were carried out as described previously [60]. FMCAK was used in these assays to be comparable to the FRET screen. Our lab previously developed and characterized a FRET tagged *Xenopus* MCAK, which has high homology with MCAK, using the same FRET pair [56]. FRET tagged *Xenopus* MCAK did not show reduced MT depolymerization activity [56]. A similar fusion of mEmerald and mCherry to human MCAK was used for analysis of MCAK conformation in human cells [32]. A mixture of 16 nM FMCAK in BRB80, 1 mM DTT, 88 mM KCl, 3 mM Mg-ATP, 0.2 mg/mL casein was added to 0.8 µM paclitaxel stabilized MTs in BRB80 with 20 µM paclitaxel and incubated for 60 min at room temperature (RT). A 3 µL aliquot was removed from the reaction at 0 min, at 30 min, and at 60 min post-protein-addition and squashed onto a coverslip. Images were acquired using a Plan Apo VC 60X/1.40 NA oil objective on a Nikon 90i microscope as described below.

For the sedimentation-based MT depolymerization assay, MTs were polymerized at a concentration of 10 µM in the presence of 1 mM β,γ-methyleneguanosine 5′triphosphate (GMPCPP, Jena Scientific, Jena, Germany) and 1 mM DTT in BRB80 for 30 min at 37 °C from a mixture of unlabeled tubulin and X-rhodamine labeled tubulin (26.5% labeled final ratio). MTs were pelleted by spinning at 45,000 rpm for 5 min at 35 °C in a TLA100 rotor in a Beckman OptimaTM MAX-TL ultracentrifuge and resuspended in BRB49 buffer. FMCAK protein (25 nM) was incubated with 1 µM microtubules for 10 min prior to fixation with 1% glutaraldehyde in BRB80 for three min. Fixed samples were diluted with 800 µL BRB80 and 100 µL of the resulting diluted sample was layered to a spindown tube loaded with 5 mL BRB80 underlaid with a 2 mL cushion of 10% glycerol in BRB80. Samples were sedimented onto poly-L-lysine coated coverslips at 12,000 rpm (22,579× *g*) in an Avanti J-25 centrifuge (Beckman Coulter, Indianapolis, IN, USA) for 1 h and post-fixed with methanol for 5 min at −20 °C. Coverslips were washed with TBS-TX and mounted on slides with Prolong Diamond. 

### 2.9. High Throughput Image-Based MT Depolymerization Assay

For high throughput analysis of the MT depolymerization activity of FMCAK, we developed a 96-well format assay. All incubations were performed in a final reaction volume of 50 µL. Biotin-labeled-BSA (Invitrogen) at 100 µg/mL in BRB20 + 100 mM KCl was incubated in BD tissue culture-treated clear flat bottomed/black walled plates (FALCON) for 30 min at RT. Wells were then washed two times with 50 µL of Tris wash buffer (40 mM Tris pH 8.0, 1 mM MgCl_2_, 1 mM EGTA, 100 mM KCl) followed by a third wash in 50 µL of BRB20 + 100 mM KCl. 50 µg/mL Neutravidin (Invitrogen) in BRB20 was added to the plate and incubated for 15 min at RT to create a biotin-BSA/neutravidin protein linkage [61]. Wells were washed two times with 100 µL of Tris wash buffer followed by a third wash in 100 µL of BRB49 buffer + 100 mM KCl. Polymerized, paclitaxel-stabilized MTs (0.2 µM) in BRB49 containing 20 µM paclitaxel were added to the wells and incubated for 30 min at RT. MTs that did not bind were removed by washing two times in 150 µL BRB49 buffer containing 20 µM paclitaxel and then 50 µL of BRB49 buffer containing 20 µM paclitaxel was added to create a well volume of 200 µL. To assay MT depolymerization, 50 µL (1/5 final well volume) of purified FMCAK was added to experimental wells at 5X concentration (15 nM) to achieve a final concentration of 3 nM in enzyme buffer (BRB80, 100 mM KCl, 2 mM Mg-ATP, 0.2 mg/mL casein, 1 mM DTT). As a positive control for inhibition of MT depolymerization activity, FMCAK was pre-incubated at a concentration of 30 nM for 15 min with either a *Xenopus* anti-MCAK NT antibody at 100 µg/mL in enzyme buffer, or FMCAK was pre-phosphorylated with Aurora B kinase at 3 nM in enzyme buffer [56] and then diluted into the assay mixture. FMCAK (alone or pre-treated with antibody or Aurora B) was added to wells containing stabilized MTs at a final molar ratio of 1:66 (MCAK:MTs). Images were captured at 0 min (before MCAK addition), and at 30 min and 60 min post-enzyme-addition as outlined below.

### 2.10. Image Acquisition and Quantification for MT Depolymerization Assay

For visual MT depolymerization assays, wide field fluorescence image acquisition utilized a Nikon Eclipse 90i fluorescence microscope with a Plan Apo VC 60X/1.40 NA oil objective and a CoolSnap HQ CCD camera (Photometrics, Tuscon, AZ, USA). Metamorph was used to control the camera, shutter, and emission/excitation filters. All samples were previewed before imaging using the show live image feature to acquire an appropriate field of view of labeled MTs. Digital images were acquired at equivalent 200 ms exposures at various times post-enzyme-addition.

High throughput images were acquired using the BD pathway 855 microscope with the UAPO/340 40X Olympus 0.9 NA dry objective with a coverslip thickness dial setting of 0.17. A BD macro was used to collect the images (rhodamine dye, 2X2 binning, 2X2 montage, 200 ms exposure, gain = 85, offset = 255, a 12 bit (1–4094 grayscale levels) dynamic range, no auto focus, and raster well movements). Dynamic range is the maximum achievable signal divided by the signal noise of both dark and read noise representing the range of grayscale levels detectable in an image. Raster well movements cause images to be taken in a sweeping zig-zag pattern for efficient total plate readings rather than using conventional left-to-right reading. Well A1, which was a buffer control, was used to set imaging focus and well depth parameters using the laser manual focusing feature for determining appropriate z-positioning.

Acquired images (a 2X2 montage for each condition) were analyzed for MT polymer amounts using a custom-generated algorithm with FIJI. Images, in tiff format, were opened in FIJI and then processed with the “sharpen” function to create more distinguished objects followed by the “find edges” function to create outlined objects. Outlined objects, highlighted in white against a black background, were set to a binary threshold making particles to be analyzed black and the corresponding background white. To filter out tubulin aggregates or other non-MT structures, a size threshold was set to 0.0015–0.1 pixel units, and circularity was set to 0–0.7. Size thresholding removed particles too small or large for the expected MT size, and circularity removed particles too circular or clumpy from corresponding images. From these established criteria, the particles were analyzed, and a summary results window of the analyzed particles was produced where particle count, particle average size, mean intensity value, and percentage coverage of the field of view is displayed. From these data, particle number (number of MTs) and average particle size (average MT pixel area) were transferred to Excel sheet for data analysis. The MT number and MT area are multiplied together to produce MT polymer levels for the image analyzed.

Images for the sedimentation-based MT depolymerization assay were acquired using the Nikon Eclipse 90i (Nikon, Tokyo, Japan) fluorescence microscope with a Plan Apo VC 60X/1.40 NA oil objective as described above. Five images per condition were collected and analyzed by an algorithm in FIJI to automatically detect and measure MT polymer amount and length. Images, in tiff format, were opened in FIJI and then processed with the “sharpen” function to create more distinguished objects followed by the “find edges” function to create outlined objects. Outlined objects, highlighted in white against a black background, were set to a binary threshold making particles to be analyzed white and the corresponding background black. To filter out tubulin aggregates and other non-MT structures, a size threshold was set to 110-infinity pixel units, and circularity was set to 0–0.7. Size thresholding removed particles too small for the expected MT size, and circularity removed particles too circular or clumpy from corresponding images. From these established criteria, the particles were analyzed, and a summary results window of the analyzed particles was produced where particle counts and particle perimeter lengths were displayed. Particle perimeter was converted to particle length (MT length) by the formula Length = (Perimeter − 12)/2 where 12 is double the average width of a MT post-processing. From these data, particle number (number of MTs) and particle length (MT length) were transferred to an Excel sheet for data analysis. The length of all MTs in a condition were added together to produce a total polymer length, which was normalized to a corresponding FCP control condition.

### 2.11. Förster Resonance Energy Transfer (FRET) Assays

Characterization of FRET activity of FMCAK was carried out as described previously [56]. Equal molar amounts (150 nM) of purified FMCAK, FCP, mCitrine, mCerulean, mCitrine + mCerulean combined, or FMCAK + Aurora B diluted in BRB49, 20 mM KCl, 2 mM Mg-ATP were excited at 433 nm, scanned at 300 nm/min, and emission was collected from 445–600 nm using a Perkin Elmer LS 50B spectrometer. The acceptor emission of mCitrine (excitation 485 nm, emission 535 nm) was used as an internal control for protein concentration. The FRET ratio (I_F_/I_D_) was calculated by dividing the fluorescence emission value at 525 nm (I_F_ of mCitrine) by the mCerulean donor emission at 475 nm (I_D_ of mCerulean) [56]. Recordings and measurements were performed in triplicate, and FRET ratios are reported as averages of those three readings.

For high throughput FRET analysis of FMCAK, the assay was adapted into a 96-well plate format using the ThermoScientific Appliskan hardware [56]. Equal molar amounts (50 nM) of purified FMCAK, FCP, FMCAK + antibody, or FMCAK + Aurora B in BRB49 containing 2 mM MgATP and 0.2 mg/mL casein were added to individual wells of black 96-well, half-area, microtiter plates (Costar^®^, Glendale, AZ, USA) in a volume of 60 µL. The indicated proteins were excited using the FRET filter pair (430/10 nm and 535/20 nm) and an mCerulean filter pair (430/10 nm and 480/10 nm), and emission was read using the 535/20 nm and 480/10 nm bandwidth of those filters, respectively. The FRET ratio was determined by measuring the fluorescence emission with the FRET filter pair (I_F_) over the emission with the mCerulean filter pair (I_D_) and was background corrected for buffer alone. Controls for inhibition were identical to those used for the MT depolymerization assays as described above. All reported FRET ratios are the average from three independent readings. 

For adaptation of the FRET assay into 384-well format, equal molar amounts (50 nM) of purified FMCAK, FCP, or FMCAK + antibody in BRB49 with 2 mM MgATP and 0.2 mg/mL casein were added to individual wells of black 384-well, microtiter plates (Nunc, ThermoFisher) in a volume of 30 µL. FRET analysis was carried out in a Synergy H1 fluorescent microtiter plate reader (Agilent BioTek, Santa Clara, CA, USA) in which the following excitation/emission readers were taken: mCerulean (Ex at 433 nm, Em at 475 nm); mCitrine (Ex at 485 nm, Em at 525 nm), and FRET (Ex at 433 nm, Em at 525 nm). The FRET ratio was determined by measuring the fluorescence emission of the FRET ex/em (I_F_) over the emission with the mCerulean ex/em (I_D_) and was background corrected for buffer alone. Controls for inhibition were as in the 96-well FRET assay. 

### 2.12. Calculation of Z′ Score for Each Assay

To assess the utility of the MT depolymerization assay or the FRET assay for HTS, we calculated a Z′ score [62] for each assay in which Z′ = 1 − [3 (SD of PC + SD of NC)/│mean of PC − mean of NC│] where PC = positive control and NC = negative control. To calculate the Z′ for the imaging assay, we used the MT polymer amount of the MTs + FMCAK at 60 min for the positive control and the MT polymer amount of the MTs + buffer at 60 min for the negative control. For the FRET assay, the FRET ratio of FCP was used as a positive control, and the FRET ratio of the mCit + mCer combined was used as a negative control. To determine the sensitivity of each assay for detecting inhibitors, we calculated a Z′ with FMCAK in the presence and absence of known inhibitors (anti-MCAK or pre-phosphorylation with Aurora B kinase). Under these conditions FMCAK + buffer is the positive control, and FMCAK with Aurora B or antibody is the negative control.

### 2.13. Small Molecule Compound Library

Small chemical compounds from the ChemBridge 50K library were obtained from the Indiana University School of Medicine Chemical Genomics Core. Compounds were supplied at a concentration of 25 µM pre-frozen in a 2.5% DMSO–water solution in 96-well polypropylene plates. Plates were stored at −20 °C until use. Plates were thawed at RT and centrifuged using a StarLab plate centrifuge (USA Scientific, Ocala, FL, USA) before use. Columns 1 and 12 contained 2.5% DMSO–water solution without compounds so that these wells could be used for assay controls.

### 2.14. Pilot Screening Using Both Imaging and FRET Assays

All compounds were screened in side-by-side assays using both image-based and FRET-based assays such that the compounds were thawed only once, and their effects were compared between assays performed on the same day. Taxol-stabilized MTs were prepared and bound to the plate as described above and imaged at time 0 min in the absence of FMCAK. While the bound MTs were imaged, the high throughput FRET assay was set up. Compound plates to be used for that day were thawed at RT for 10 min and centrifuged. The following plate set-up was used: the FCP (35 µL, 80 nM in BRB80, 63 mM KCl, 1.6 mM DTT, 4.8 mM Mg-ATP, 0.32 mg/mL casein) was added to column one wells (A1-H1). FMCAK was added at the same concentration and volume to the first four wells of column 12 (A12-D12). FMCAK that was pre-incubated with antibody was added to the bottom four wells of column 12 (E12-H12). The rest of the wells (Columns 2 through 11) contained FMCAK. The compound solutions (20 µL) were removed from the compound plates, added to the assay plates with protein (Columns 2 through 11), and mixed such that the final volume of the assay was 55 µL with a final concentration of 9.08 µM compound, 0.91% DMSO, 50 nM protein, 40 mM KCl, 1 mM DTT, 3 mM Mg-ATP, and 0.2 mg/mL casein. Plates were incubated at RT for 15 min and then read as previously described for FRET.

While the FRET plates were being read in the Appliskan (Thermo Scientific), the imaging assay was prepared. FMCAK was mixed with enzyme buffer or antibody and pre-incubated for 15 min by adding it to the compound plates at RT (which had 30 µL of compound remaining in each well). The plate set-up involved adding 30 µL of either protein sample to compound/control wells and 30 µL of enzyme buffer to column 1 control wells, and mixed to have a final volume of 60 µL at 15 nM protein. After the pre-incubation, the protein–compound mixtures were added to imaging plates with MTs bound (250 µL final volume and 3 nM of final protein). Images were acquired at 30 min and 60 min post-protein-addition as detailed above.

The second, 1280 compound screen was carried out in duplicate using the 384-well FRET assay described above. Compounds were screened side-by-side in both replicates such that each compound was only thawed once. Compounds were thawed at RT for 10 min and centrifuged. The following plate setup was used: FCP (19.1 µL, 80 nM in BRB80, 63 mM KCl, 1.6 mM DTT, 3.2 mM MgATP, 0.32 mg/mL casein) was added to wells B1-P1. FMCAK was added at the same concentration and volume to wells B24-H24. FMCAK that was pre-incubated with antibody was added to wells I24-P24. The rest of the wells (Columns 2 through 23) contained FMCAK. The compound solutions (10.9 µL) were removed from the compound plates and added to the assay plates with protein (Columns 2 through 23) and mixed such that the final volume of the assay was 30 µL with a final concentration of 9.08 µM compound, 0.91% DMSO, 50 nM protein, 40 mM KCl, 1 mM DTT, 2 mM Mg-ATP, and 0.2 mg/mL casein. Plates were incubated at RT for 15 min and then read as previously described for FRET.

### 2.15. Re-Screening and Secondary Confirmation Assays

For the image-based assay, wells that had greater than 50% of polymer levels (amount of polymer at 60 min/ amount of polymer at 0 min) were considered inhibitors and those that had less than 10% polymer levels were considered as activators. Wells in which the FMCAK FRET ratio increased or decreased by 15% or more were considered activators or inhibitors respectively.

Compounds that scored in either assay in the pilot screen initially (a total of 65 compounds) were ordered for re-screening. Assay conditions for both the FRET and image rescreen assays were identical to the pilot screen with the exception that wells A7-H11 on re-screen plate 2 only had H_2_O with DMSO at 2.5%. During the re-screen, we also performed a counter screen to identify any compounds that affected the fluorescence properties of FCP and thus were likely false positives. All protein concentrations, buffers, volumes, and assay conditions were identical to those of the primary screen. Thresholds for hits were the same as used for the primary screens. The three identified inhibitors (2021-4 C4, 2030-1 B4, and 2042 H9) had their chemical formulas confirmed by ESI mass spectrometry by the IU Mass Spectrometry Facility using the LQT-Orbitrap XL (Thermo Scientific, Waltham, MA, USA).

### 2.16. Colony Formation Assays

Cells were seeded in 24-well plates at a density of 100 cells per well. After incubating for 24 h, cells were treated with DMSO or the indicated concentrations of MCAK inhibitors and allowed to grow for nine days. The cells were rinsed in pre-warmed PBS and then fixed in a solution of 4% formaldehyde in PBS for 30 min. The cells were rinsed with PBS and incubated at RT in 0.5% *w*/*v* of crystal violet in PBS for 1 h. Cells were rinsed with deionized water and left to dry overnight before imaging. The following day, whole well montages were imaged on the Lionheart FX Automated Microscope^®^ (Agilent BioTek, Santa Clara, CA, USA) using a 4X phase objective (4X 0.13 Plan Fluorite phase) and images were stitched together to create a single well image. Image files were opened in FIJI software, and colonies with greater than 50 cells were manually counted. Colony counts from each individual experiment were normalized to DMSO controls and plotted using the GraphPad Prism software (GraphPad Software Inc.) for at least three individual experiments. Normalized values were compared to a theoretical value of 1.0 (100% of DMSO control) using a one sample *t*-test. 

### 2.17. Quantification and Statistical Analysis

Unless otherwise noted, all experiments represent averages of at least three biological repeats ± the standard deviation. All data was analyzed and plotted using Excel (Microsoft) and GraphPad Prism (Graphpad Software Inc.). For MTT experiments, OD values were normalized to a DMSO control and plotted in GraphPad Prism. IC_50_ values were calculated using the log_10_ (inhibitor) vs. response (three parameters) analysis option. This option was most appropriate for the assay as four parameter fits require more data points in order to calculate a proper curve. For immunofluorescence experiments, at least 100 cells for each condition were counted per biological repeat. Student’s *t*-tests were used to determine *p*-values for all assays scoring either micronuclei or lagging chromosome. For clonogenic survival, the average number of colonies per condition was normalized against control conditions and plotted to generate a percentage of the control condition. Averages of the repeats were compared to a theoretical value of 100% using a one sample *t*-test. For all experiments, * *p* < 0.05, ** *p* < 0.01, *** *p* < 0.001, **** *p* < 0.0001.

## 3. Results

### 3.1. High MCAK Expression in Breast Cancer Is Associated with Poor Prognosis

Previous studies [41,42,50] showed that MCAK expression is higher in multiple types of cancer tissue relative to normal tissue. To explore this in the context of breast cancer, we examined MCAK expression in the TCGA database and found that MCAK expression was significantly higher in cancer tissue relative to normal breast tissue (Figure 1A). When stratifying the data relative to receptor status, MCAK expression was highest in triple-negative breast cancer (TNBC) (Figure 1B), suggesting that it may be correlated with more lethal subtypes. MCAK expression was upregulated across PAM50 subtypes and was highest in the basal subtype (Appendix A). Consistent with these data, MCAK expression was associated with worse overall survival (Figure 1C) and with poorer recurrence-free survival (Figure 1D). Surprisingly, MCAK expression was higher in tumors with pathological complete response (Appendix A) relative to those with residual disease. This observation suggests that patients with high MCAK expression initially respond to treatment but then relapse, leading to the poor overall survival. These data mirror patients with TNBC, who initially respond well to chemotherapy but eventually relapse with resistant disease [51]. Overall, MCAK expression was found to be higher in more aggressive breast cancers and associated with poorer patient outcomes.

### 3.2. Loss of MCAK Induces Aneuploidy in Taxane-Resistant Cells

MCAK is a MT depolymerase that is critical for preventing/correcting aberrant MT attachments to kinetochores, which otherwise lead to CIN and aneuploidy in daughter cells [26,27,29]. Front-line breast cancer treatment usually involves taxanes, which alter MT dynamics [14] and induce aneuploidy in breast cancer cells [17,63]. We therefore rationalized that loss of MCAK may sensitize cells to taxanes. To test this idea, we measured the IC_50_ for paclitaxel in a series of tumor-derived versus non-tumor-derived cell lines. We found an approximately two-to-five-fold decrease in the IC_50_ in tumor-derived lines (MDA-MB-231, HeLa, U2OS) whereas there was no change in the non-tumor-derived cell lines (MCF-10A, RPE-1) (Figure 2A,B and Appendix A). To ask if this difference in sensitivity was due to a synergistic increase in aneuploidy, we knocked down MCAK, treated cells with 2.5 nM or 5 nM paclitaxel, and then incubated cells for 24 h in cytochalasin B to induce cytokinesis failure, resulting in an accumulation of binucleate cells. Binucleate cells containing micronuclei are indicative of chromosome missegregation events (Figure 2C). Either MCAK knockdown or treatment with paclitaxel caused a two-to-three-fold increase in the percentage of binucleate cells containing micronuclei, but there was no synergistic increase in the percentage of cells with micronuclei when MCAK knockdown cells were combined with paclitaxel addition (Figure 2D). However, MCAK knockdown resulted in an increase in the percentage of binucleate cells containing micronuclei in a paclitaxel-resistant version of MDA-MB-231 cells (Figure 2E and Appendix A), suggesting that MCAK inhibition may provide a mechanism to induce aneuploidy in taxane-resistant tumor cells. Given that taxane resistance is a major factor in the poor prognosis of breast cancer patients, especially those with TNBC, targeting MCAK may provide a valuable new therapeutic strategy.

### 3.3. Identification of Three Putative MCAK Inhibitors

Our results above support the idea that MCAK inhibitors may be beneficial as taxane sensitizers, inducers of aneuploidy, or as secondary agents in cases of taxane resistance in TNBC. We therefore adapted two assays for MCAK inhibition into 96-well plate format to use for inhibitor screening. Previous studies showed that MCAK could depolymerize paclitaxel-stabilized MTs in an ATP-dependent manner either in solution or when bound to glass [21]. Wells of a 96-well plate were coated with biotin-BSA followed by Neutravidin to create the binding surface (Figure 3A). X-rhodamine-biotin-MTs were polymerized with GTP, stabilized by the addition of paclitaxel, and then bound to the wells through a biotin–Neutravidin linkage. MT images were recorded at *t* = 0 min before enzyme addition. MCAK in buffer containing ATP was then added to the wells and imaged at 20 min intervals post-enzyme-addition. To measure the activity of MCAK, we developed a custom algorithm using FIJI to automatically measure length and number of MTs in the image (Figure 3A). The sum of the MT lengths was calculated to reflect the total amount of polymer in the imaged region. Using this assay, we were able to measure MCAK MT depolymerization activity in a time-dependent fashion (Appendix A). MCAK activity can be inhibited by phosphorylation with Aurora B kinase or by addition of an inhibitory antibody [64,65,66,67], and both inhibitors blocked MCAK activity in our assay (Appendix A). These results showed that our assay can detect MCAK inhibition of MT depolymerization activity. The Z′ of this assay from our optimization experiments in multi-plate format was 0.71 with a CV of 20%, making it potentially suitable, but not ideal, for screening. Previously, we developed a FRET reporter of *Xenopus* MCAK, which showed high FRET when MCAK was in a closed, active conformation and reduced FRET when MCAK was in an inhibited, open conformation [56]. We made a similar reporter for hMCAK in which an mCerulean donor was fused to the C-terminus of MCAK, and an mCitrine acceptor was fused to the N-terminus of MCAK and used this to measure FRET (Figure 3B). Upon excitation at 525 nm, FMCAK exhibited FRET with a FRET ratio of 1.64 ± 0.02 (Appendix A). Incubation with Aurora B kinase, which inhibited MCAK MT depolymerization activity, reduced the FRET ratio to 1.1 ± 0.01, and incubation with anti-MCAK antibodies reduced the FRET ratio to 1.34 ± 0.03 (Appendix A). Because this assay was already being carried out in a 96-well plate format, we used our data from multiple experiments to estimate a Z′ for this assay of >0.9 with a CV of 2%, suggesting it had potential utility for high throughput screening. 

We completed a pilot screen with 3000 compounds from the Chembridge 50K library in which each compound was simultaneously tested in both assays. On a given day, we thawed one 96-well plate of compounds and then pre-incubated the compounds with FMCAK followed by dilution for either the image-based assay or the FRET assay. This allowed us to test identical compounds on the same fraction of FMCAK congruently. For the image-based assay, a typical plate result is shown in Figure 3C, which shows the amount of MT polymer remaining in each well at 60 min post-FMCAK-addition. The FRET control protein (FCP, cyan) in which mCitrine was fused directly with mCerulean [56,68] had no effect on MTs, so the amount of MT polymer is high. The addition of FMCAK (green) resulted in MT depolymerization and lowered the amount of MT polymer. This effect could be reduced by the addition of an anti-MCAK antibody, which inhibits FMCAK activity (red) and resulted in MT polymer levels similar to those with FCP. The addition of most compounds had no effect on the amount of MT polymer (blue dots between red and purple dashed lines); however, we did identify compounds that inhibit MT depolymerization activity (red circle) and occasionally found compounds that stimulated MT depolymerization activity (purple circle). For the FRET assay, a typical result is shown in Figure 3D. In this assay the FCP (cyan) served as a positive control for FRET because this protein has the two FRET proteins (mCit/mCer) fused together without any intervening MCAK and thus has very high FRET. FMCAK (green) had a FRET ratio of approximately 1.65, which could be reduced by 15% in the presence of inhibitory anti-MCAK antibodies (red). The FRET assay could identify compounds that either inhibited (red circles) or activated (purple circles) the FMCAK protein. From the 3000-compound pilot screen we identified a total of 27 hits in the image assay and 38 hits in the FRET assay with 6 compounds scoring in both assays (Appendix A). We then rescreened candidate inhibitors via hit confirmation assays, which yielded 14 hits in the image assay and only 2 hits in the FRET assay, both of which scored in the image-based assay. We also conducted a counter screen in which the FCP was used to identify any compounds that non-specifically reduced or increased the fluorescence of the fluorophores themselves. None of the 14 compounds identified in the pilot screen scored in this counter screen, suggesting that the identified inhibitors are not acting as non-specific agents.

There were both advantages and disadvantages in the two assays that needed to be considered. The image-based assay had a higher hit rate, leading to more compounds identified in this pilot screen. However, this hit rate could lead to more false positives since this assay had a lower initial Z′ (0.71). The image-based assay is also limited by the time it takes to read a single plate (~12.5 min for a complete read), such that to assay more than one plate at a time (which we routinely do), we cannot set up multiple plates simultaneously. In addition, the binding of the MTs to the wells appears to be a major contributor to the variability and the high CV. Another issue with the imaging assay is that there were distortions in the imaging unless the wells contain at least 200 µL of buffer. This means that, with a library stock compound concentration of 25 µM, we are only able to test compounds at a final concentration of 2 µM in the primary assay. Due to these limitations, we concluded that the image-based assay is not suitable for development as a HTS assay but would be a powerful secondary screening orthogonal assay, in which there is a more limited processing of compounds. In contrast to the image-based assay, the FRET assay is very rapid to carry out, uses far fewer reagents (significantly reducing cost), and is readily adaptable to a 384-well plate format. Our initial hit rate with the FRET assay was somewhat high (1.3%) but the rescreen confirmation assay returned very few compounds such that the final hit rate was only 0.05%. Of note is the finding that both hits in the FRET assay scored in the image assay, suggesting that this assay identifies compounds that also score in an orthogonal assay. 

To further develop the FRET assay for screening, we optimized conditions for this assay in 384-well format and carried out a second pilot screen of 1280 compounds. Since the initial hit rate in the FRET assay was higher than desired, we completed the FRET assay in duplicate, where we tested each compound on two individual plates that were assayed simultaneously. A typical result from a single plate is shown in Figure 3E. The Z′ calculated from controls on this screen was 0.8 with a CV of 5%, suggesting that it is suitable as a HTS assay. From this 1280 compound screen we identified six hits on each replicate plate, of which three compounds were overlapping between the two replicate plates (Appendix A), yielding a hit rate of 0.2%, a manageable value for future large-scale screens. In the hit confirmation assay, one out of the three compounds scored positive. Similar to the hits obtained from the 3000-compound screen, no compounds were positive against the FCP used as a counter screen (Appendix A). Between the two pilot screens of 4280 compounds, we identified three confirmed hits that were potential MCAK inhibitors, 2021-4 C4 (C4), 2030-1 B4 (B4), and 2042 H9 (H9) (Figure 3F). 

To characterize the relative potency of the identified hits in inhibiting MCAK-mediated MT depolymerization, we pre-incubated 50 nM FMCAK with 100 nM, 1 µM, or 10 µM of each inhibitor for 10 min and then added this to an equivalent volume of 2 µM of GMPCPP stabilized, X-rhodamine labeled MTs (for final inhibitor concentrations of 50 nM, 500 nM, and 5 µM respectively). This mixture was incubated for 10 min before fixing and sedimenting onto coverslips to view by immunofluorescence microscopy (Figure 4A). The total length of MT polymer was quantified and normalized to an FCP control condition as an assessment of activity (Figure 4B–D). The addition of FMCAK to MTs caused 90% of the MTs to be depolymerized. Pre-incubation of FMCAK with an anti-MCAK antibody caused an approximate two-fold reduction in FMCAK activity, resulting in more MTs remaining on the coverslip. For compounds B4 and H9, only the 5 µM concentration of drug inhibited activity, whereas compound C4 inhibited FMCAK activity at both 500 nM and 5 µM concentrations. It is worth noting that compound H9 was initially identified solely through the FRET-based assay, suggesting that the FRET assay can identify inhibitors that also modulate MCAK-mediated MT depolymerization activity. Collectively, these data show that we identified three compounds that inhibit MCAK activity in vitro. 

### 3.4. MCAK Inhibitors Induce Aneuploidy in Taxane-Resistant Cells

To ask whether the putative MCAK inhibitors could inhibit MCAK activity in cells, we assessed their ability to induce chromosome missegregation. We treated MDA-MB-231 cells with 50 µM of the indicated inhibitor for 24 h in the presence of 3 µM Cytochalasin B to induce cytokinesis failure. All three inhibitors resulted in an approximate three-fold increase in the percentage of binucleated daughter cells that had a micronucleus (Figure 5A), suggesting that the inhibitors promote the induction of aneuploidy. These compounds also resulted in the formation of micronuclei in the taxane-resistant PTXR-231 cell line [52] (Figure 5B), providing additional support for the idea that MCAK inhibitors are effective in taxane-resistant cells. To test the specificity of the inhibitors, we assayed the percentage of anaphase cells with lagging chromosomes after doxycycline-inducible, CRISPR-mediated knockout of MCAK in HeLa cells [53,54]. Upon doxycycline treatment, we estimated that about 60% of the cells had MCAK knocked out as evidenced by loss of MCAK staining in immunofluorescence (Figure 5C). We then scored the percentage of cells that had lagging chromosomes in anaphase. The addition of the inhibitors caused an approximate two-fold increase in the percentage of cells with lagging chromosomes relative to control DMSO addition (Figure 5D). MCAK knockout plus control DMSO also caused an approximate two-fold increase in the percentage of cells with lagging chromosomes. Importantly, the inhibitors did not cause a further increase in the percentage of cells with lagging chromosomes. We observed similar results when the inhibitors B4 and C4 were compared to MCAK knockdown (Appendix A). Together, these results suggest that the lagging chromosomes phenotype is not due to off-target effects of the drugs. 

### 3.5. MCAK Inhibition Enhances Taxane Sensitivity and Reduces Colony Formation

One important finding from our initial studies was that MCAK knockdown sensitizes cells to taxanes. If MCAK inhibitors act as taxane sensitizers, they could be used in combination therapies with taxanes to allow for lower dosages of taxanes and subsequently fewer dose-limiting side effects. To ask whether treatment with the inhibitors would sensitize cells to taxanes, we measured the IC_50_ for paclitaxel in MDA-MB-231 cells in the presence of 50 µM of each of the inhibitors, a concentration sufficient to induce chromosome missegregation. While compounds B4 and H9 did not significantly alter sensitivity to paclitaxel, compound C4 caused a 2.3-fold reduction in the IC50 (Figure 6A). 

The above results show that MCAK inhibition can induce aneuploidy after a single mitotic division (Figure 5 and Appendix A), but they do not account for the longer-term consequences of chromosome missegregation through multiple divisions. To ask whether MCAK inhibition affected the ability of cells to grow into colonies and survive successive rounds of division, we seeded cells at low density and treated them with three concentrations of indicated drug and allowed nine days of growth before staining with crystal violet. MDA-MB-231 cells are highly migratory and tend to form amorphous colonies (Figure 6B). The addition of each drug reduced the average number of colonies with C4 being the most potent (Figure 6C). Similar to our findings with the aneuploidy induction assays, the MCAK inhibitors also reduced colony formation in the taxane-resistant derivative cell line PTXR-231 (Figure 6D,E). These findings suggest that MCAK inhibitors may have potential use as single agents, particularly in the case of taxane-resistant disease.

## 4. Discussion

Our work suggests that MCAK may be a prognostic marker for breast cancer outcomes as well as a good candidate for therapeutic intervention. MCAK inhibition may be beneficial as a sensitizer to taxanes, which may provide a way to reduce taxane-mediated side effects while maintaining the clinical efficacy of taxanes. Another possibility is that MCAK inhibitors may be beneficial in the context of taxane resistance, which is common in TNBC [51]. Together, our work supports the pursuit of MCAK as a potential therapeutic target.

Many tumors exhibit CIN, leading to increased levels of aneuploidy. Aneuploidy can be advantageous for tumors because it allows for heterogeneity, leading to favorable karyotypes that enhance proliferation, accelerate drug resistance, and/or reduce cell death [2,6,69,70,71,72,73,74]. In addition, aberrant chromosome segregation can result in the formation of micronuclei, inducing several downstream consequences, including chromothrypsis, stimulation of the immune response, and epigenetic changes in daughter cells [2,3,4,5,31]. Too much aneuploidy can lead to lethality due to loss of genetic material. It is postulated that taxanes may act by inducing lethal levels of aneuploidy [14]; therefore, targeting other molecules that alter aneuploidy induction may be a valuable clinical strategy.

MCAK may be an under-appreciated and much needed target for its prognostic value and potential therapeutic development. Triple-negative breast cancer carries the worst prognosis of all the subtypes, in part due to the lack of targeted therapies available for treatment. In breast cancer, MCAK/KIF2C is part of the PAM50 signature for classification of breast cancer subtypes where high expression is associated with the basal subtype [38,75]. This is consistent with our observation of MCAK expression being highest in basal tumors, which make up the majority of TNBCs. Tumors that initially responded to treatment, resulting in a pathological complete response, were found to have poorer long-term survival. This could be due to increased risk of metastasis or to disease recurrence as with other cancers [41,42]. This paradoxical association with pathological complete response and survival also mirrors the observation that TNBC patients initially respond favorably to taxanes, but often relapse with resistant forms of disease [51]. Since MCAK inhibitors appear to be effective in taxane-resistant cells, MCAK-based therapeutics may serve as valuable second-line therapies for resistant disease. MCAK expression was also found to be a global marker for poor prognosis in multiple cancers, suggesting that inhibition of MCAK may be a viable approach in other cancer types as well [37]. Many of the co-regulated genes found in this genome-wide study were general cell cycle regulators; however, inhibition of MCAK does not result in mitotic arrest or mitotic catastrophe. These data suggest MCAK inhibition may not result in the same toxicities associated with other global cell cycle regulators.

Neuropathy is a harmful, dose-limiting side effect of taxanes. One of the promises of combination therapy in cancer is the reduction of side effects by combining drugs that synergize without having overlapping toxicities. The ability to sensitize cells to taxanes with MCAK inhibition is striking in this context. Combining MCAK inhibitors with taxanes could significantly reduce these side effects, improving patient outcomes and quality of life. MCAK function has been further linked with taxane cytotoxicity through tubulin post-translational modifications that inhibit the enzyme [76]. As a mediator of DNA repair, MCAK inhibition may synergize with taxanes to increase DNA damage in the cell, as taxanes have been shown to synergize with certain DNA damaging agents [77]. Microtubules have been proposed to play a number of roles in DNA repair which are inhibited by modulating their dynamics [78]. For example, taxanes have previously been shown to inhibit DNA repair through interfering with the trafficking of DNA repair proteins [79]. MCAK inhibition and taxane treatment may therefore combine to increase DNA damage in the cell. Additionally, MCAK inhibition may combine favorably with other therapies. For instance, increased CIN, as is caused by MCAK inhibition, has been shown to trigger IL-6 signaling through cGAS-STING to increase tumor cell survival [80]. A combination of MCAK inhibitors with IL6R inhibitors may be a potential way to increase the efficacy of both drugs, while counteracting some of the pro-tumorigenic effects of CIN [80]. 

We were surprised that the combination of MCAK inhibition/knockdown with taxanes did not lead to an increase in lagging chromosomes or micronuclei. It is important to note that our assay examines micronuclei after a single division, as assessed by a failure in cytokinesis. If MCAK inhibition causes a low level of chromosome missegregation, multiple rounds of division may be needed to accumulate serious defects. It is also possible that the combination treatment resulted in complete missegregation of chromosomes without the generation of a micronucleus, which could lead to longer-term defects in cell proliferation and viability only after multiple rounds of division. The effects of MCAK inhibitors reducing clonogenic growth support the idea that MCAK inhibitors may need multiple rounds of division to cause severe defects. It is interesting that only compound C4 reproduced the sensitization to paclitaxel seen in our knockdown experiments despite not inducing significantly more micronuclei or lagging chromosomes compared to the other two inhibitors. The taxane-sensitizing effects of compound C4 appear to be MCAK specific given that C4 did not increase the number of lagging chromosomes when combined with MCAK knockdown/knockout. It was clearly the most potent inhibitor in our in vitro assays. MCAK inhibition most likely produces different effects depending upon the level of inhibition. One possibility is that C4 is impairing a recently discovered role of MCAK in DNA repair and that inhibiting this function synergizes with taxane treatment [81], an idea that we will be evaluating in the future. 

While MCAK has been examined as a putative prognostic marker for multiple cancers [36], there has been little effort focused on developing tools to probe MCAK function or that could be developed into clinical effectors. A prodrug was identified and shown to target MCAK and induce spindle defects consistent with MCAK inhibition, but this compound was non-specific and bound to multiple targets in addition to MCAK [82]. The small molecule DHTP was identified as a non-competitive inhibitor of Kinesin-13, which inhibited the ATPase activity of the Kinesin-13s, MCAK and Kif2A, with a three-to-four-fold increase in specificity for MCAK over Kif2A [83]. DHTP also inhibited MT dynamics in Drosophila cultured cells, but there was no evidence of how it affected mitosis [83]. The small molecule, UMK57, was found to reduce CIN by potentiating MCAK-mediated MT destabilization activity [84]. However, tumor cells adapted within days to the presence of UMK57 by altering Aurora-B-mediated phosphorylation of MCAK, which has been shown to downregulate MCAK activity [84]. 

Our studies provide potential new chemical tools that could be used to probe MCAK function in several systems. While these inhibitors appear to be low potency, this does not preclude them from being useful as cellular probes. Monastrol was the first kinesin family member inhibitor, targeting the Kinesin-5 protein [85]. In cells, it is used at 100 µM for the generation of monopolar spindles. While it was never developed clinically, it has been used in hundreds of studies to generate monopolar spindles and to probe Kinesin-5 as a potential therapeutic target. The discovery of Monastrol also prompted screens for additional Kinesin-5 inhibitors, and at least eight of these have been/are being utilized in dozens of clinical trials [86], demonstrating the broad utility of mitotic kinesin probes.

We currently do not know the mechanism of action of the inhibitors that we identified to know whether they are catalytic inhibitors or allosteric inhibitors. Both B4 and C4 scored as inhibitors in an assay that measured MT depolymerization activity of MCAK, as well as in an assay that is based on MCAK conformation [56]. This latter finding is more consistent with the idea that these are allosteric effectors, as previous studies have shown regions outside the catalytic domain modulate MCAK conformation [56,87]. None of our three inhibitors structurally resemble ATP and are dissimilar from each other, making it unlikely that any of them are competitively inhibiting the binding of ATP to MCAK. Because MCAK undergoes conformational changes that correlate with activity, as measured by our FRET assay, it is likely the identified drugs work as allosteric inhibitors of MCAK. Ideally, allosteric effectors to regions outside the motor domain would be more specific for MCAK relative to other kinesin family members, which share high homology in the catalytic domain. Outside the catalytic domain, MCAK shares highest homology with other members of the Kinesin-13 family, including Kif2A, which is expressed and functions in neurons [88]. While it has long been thought that MCAK functions primarily in mitotic cells, a recent study [89] revealed a new function for MCAK modulating synaptic plasticity in mice, raising the question of how targeting MCAK may affect neuronal cells. Identifying inhibitors that target MCAK and not Kif2A will be crucial if they are to be utilized either in combination with taxanes or as agents to give taxane-like effects without inducing neuropathies. In addition, understanding whether/how MCAK functions in neurons will be critical to understand whether MCAK inhibitors would induce damaging neuropathies. 

In our screen, we identified three MCAK inhibitors, albeit with different potencies in reducing clonogenic growth. C4 could serve as a starting point for chemical optimization, as it is clearly the most potent in inhibiting MCAK function, sensitizing cells to taxanes, and reducing clonogenic growth. It is possible that C4 is simply the most cell permeable or the most stable, since all three compounds had similar efficacy in short-term assays associated with chromosome missegregation and the formation of micronuclei. It is also possible that C4 has off-target effects that only appear in long-term assays, such as the colony formation assay. An alternative strategy to identify improved MCAK inhibitors would be to take advantage of the optimized FRET-based assay and screen a library enriched for bioactive compounds with drug-like properties, such as the Lopac1280 (Library of Pharmacologically Active Compounds, Sigma Aldrich) collection of known drugs and bioactives or the Microsource Spectrum collection of 2400 compounds (Microsource Discovery Systems Inc., Gaylordsville, CT, USA). Either approach has the potential to identify and develop additional tools to assess the therapeutic usefulness of MCAK and to identify which aspects of MCAK function (regulation of cellular MT dynamics, inhibition of CIN, modulation of DNA damage repair, or cell migration) are most important for tumor cell survival. Future identification of more potent, more bioavailable compounds will allow us to assess the potency of these drugs in vivo, as well as their effects, if any, on peripheral nerves.

## 5. Conclusions

Collectively, we show that MCAK has potential clinical applications as both a biomarker and as a therapeutic target. MCAK expression is significantly correlated with more aggressive subtypes of breast cancer. It is also associated with several metrics of poor prognosis including overall survival and distant metastasis-free survival. We show that MCAK loss or sufficient inhibition can sensitize cells to taxanes, which may be valuable in improving patient quality of life and outcomes through reducing the incidence of dose-limiting toxicities. Our finding that MCAK inhibition reduces the clonogenic survival of taxane-resistant cells is notable in the context of TNBC, as taxane resistance frequently develops and there are few other treatment options available. Furthering our understanding of MCAK function in cancer, as well as the development of more potent inhibitors, are likely to be important steps forward in the treatment of TNBC.

## Figures and Tables

**Figure 1 cancers-15-03309-f001:**
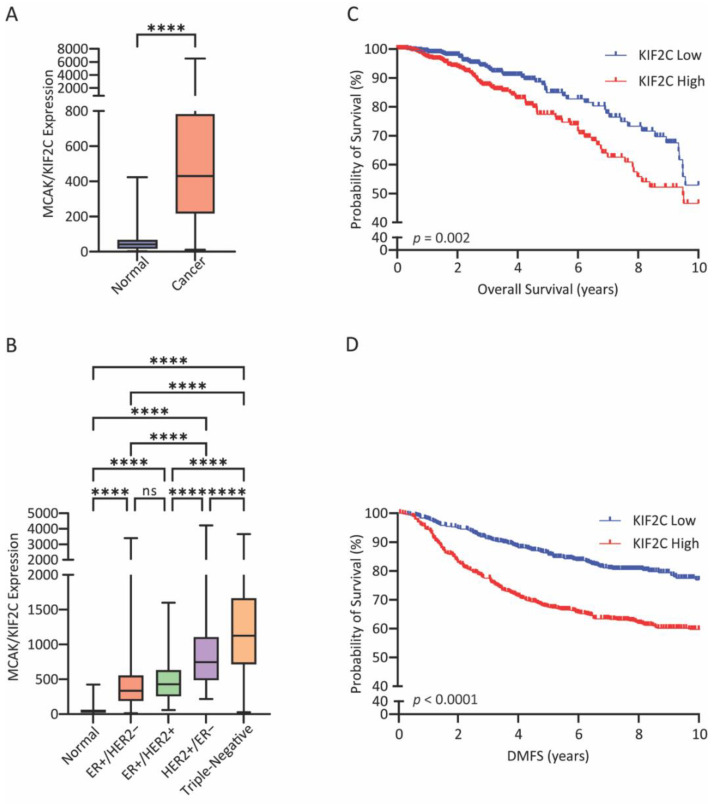
MCAK overexpression correlates with poor prognoses. (**A**) MCAK/KIF2C expression data from the publicly available The Cancer Genome Atlas Breast Cancer (TCGA BC) dataset from normal (Normal N = 112) and breast-tumor-derived samples (Cancer N = 1100) plotted as a bar-and-whisker plot where the box represents the upper and lower quartiles, and the whiskers represent the upper and lower range of values. Samples were compared with a Student’s *t*-test. **** *p* < 0.0001, ns, not significant. (**B**) MCAK/KIF2C expression data from TCGA BC dataset from breast cancer tumors of different receptor status (Normal N = 112, ER + /HER2− N = 594, ER + /HER2+ N = 123, HER2 + /ER− N = 37, Triple-Negative N = 148) plotted as bar-and-whisker. ANOVA with Tukey’s post hoc test for significance was used for comparison. **** *p* < 0.0001. (**C**) Kaplan–Meier plot of breast cancer overall survival in the TCGA BC cohort. MCAK/KIF2C expression levels were stratified with a median split into high (N = 550) and low (N = 550) expression. Log rank testing was used to compare survival for the different groups with the *p* value indicated. (**D**) Kaplan–Meier plot of breast cancer distant metastasis-free survival (DMFS) in the GSE47561 cohort. MCAK/KIF2C expression levels were stratified with a median split into high (N = 560) and low (N = 561) expression. Log rank testing was used to compare survival for the different groups with the *p* value indicated.

**Figure 2 cancers-15-03309-f002:**
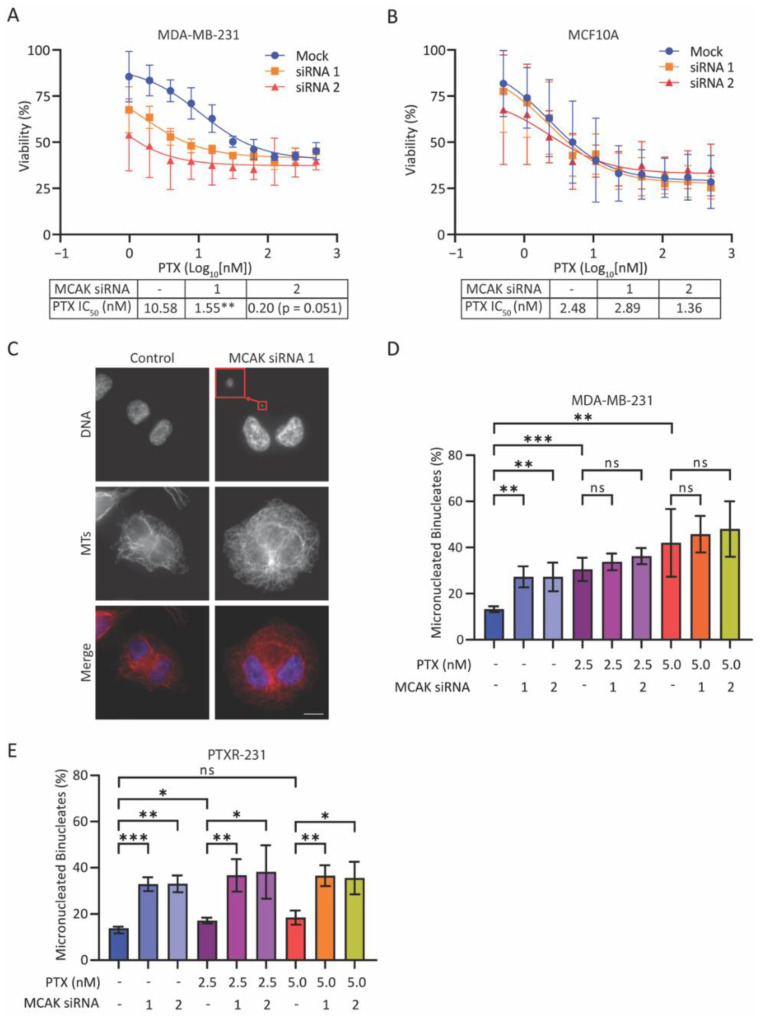
MCAK loss sensitizes cancer cells to taxanes and induces aneuploidy in taxane-resistant cells. (**A**,**B**) Dose response curves and associated IC_50_ values of paclitaxel in MDA-MB-231 (**A**) or MCF10A (**B**) cells from three and five independent experiments, respectively, in which duplicate values were averaged and normalized versus control DMSO treatment. (**C**) Representative images of binucleate MDA-MB-231 cells stained for MTs (red) and for DNA (blue). Binucleate cells were identified by the proximity and mirrored orientation of their nuclei. The enlarged red box indicates a micronucleus. Scale bar = 10 µm. (**D**,**E**) Percentage of binucleate cells with a micronucleus in MDA-MB-231 cells (**D**) (n = 4) or the taxane-resistant daughter line PTXR-231 (**E**) (n = 4). Values are plotted as the mean ± SD for each condition. Conditions were compared with a Student’s *t*-test. ns = not significant, * *p* < 0.05, ** *p* < 0.01, *** *p* < 0.001.

**Figure 3 cancers-15-03309-f003:**
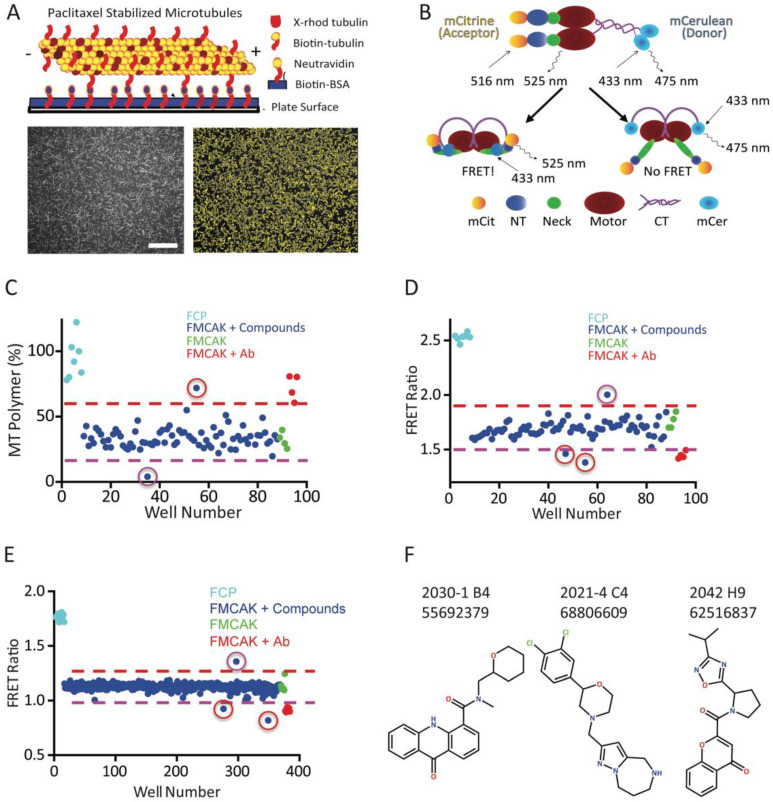
Identification of putative MCAK inhibitors. (**A**) Cartoon schematic and representative images of MT depolymerization assay. Lower left shows a representative image of the FCP control condition. Lower right shows the same image with the MTs identified by the FIJI algorithm outlined in yellow. Scale bar = 5 µm. (**B**) Schematic of MCAK FRET construct in which mCitrine (mCit) is fused to the N-terminus (NT) and mCerulean (mCer) is fused to the C-terminus (CT) of MCAK. Dimeric MCAK is in the closed active state in which FRET will occur (bottom left) or the open inactive state in which the FRET signal is greatly reduced (bottom right). (**C**) Sample data of image-based MT depolymerization assay from a single 96-well plate with the data points colored as labeled. The compound with a red circle is a putative inhibitor, whereas the compound with a purple circle is a putative activator. (**D**) Sample data of FRET assay from a single 96-well plate with the data colored as labeled. Two compounds with red circles are putative inhibitors whereas the compound with a purple circle is a putative activator. (**E**) Sample data of further optimized FRET assay from a single 384-well plate with the data as labeled. Two compounds with red circles are putative inhibitors whereas the compound with a purple circle is a putative activator. Note that the difference in Y-axis scaling relative to (**D**) is due to the use of a spectral-based plate reader in the assay in (**E**). (**F**) Structure of the three hits identified from the small-scale screens. Created with Chemdraw.

**Figure 4 cancers-15-03309-f004:**
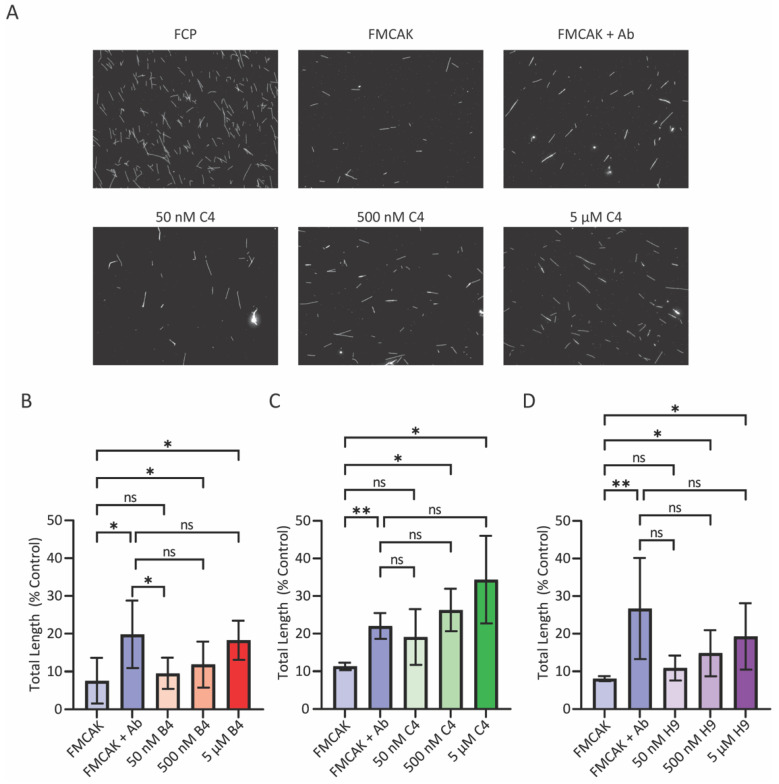
MCAK inhibitors exhibit dose-dependent in vitro potency. (**A**) Representative images from the MT depolymerization assay. Scale bar = 10 µm. (**B**–**D**) Five images per replicate per condition were taken and analyzed for total polymer length relative to the FCP control condition. The total polymer length in each condition was normalized to a percentage of FCP control condition in experiments with B4 (**B**), C4 (**C**), and H9 (**D**) drugs. Conditions were compared with a Student’s *t*-test. ns = not significant, * *p* < 0.05, ** *p* < 0.01.

**Figure 5 cancers-15-03309-f005:**
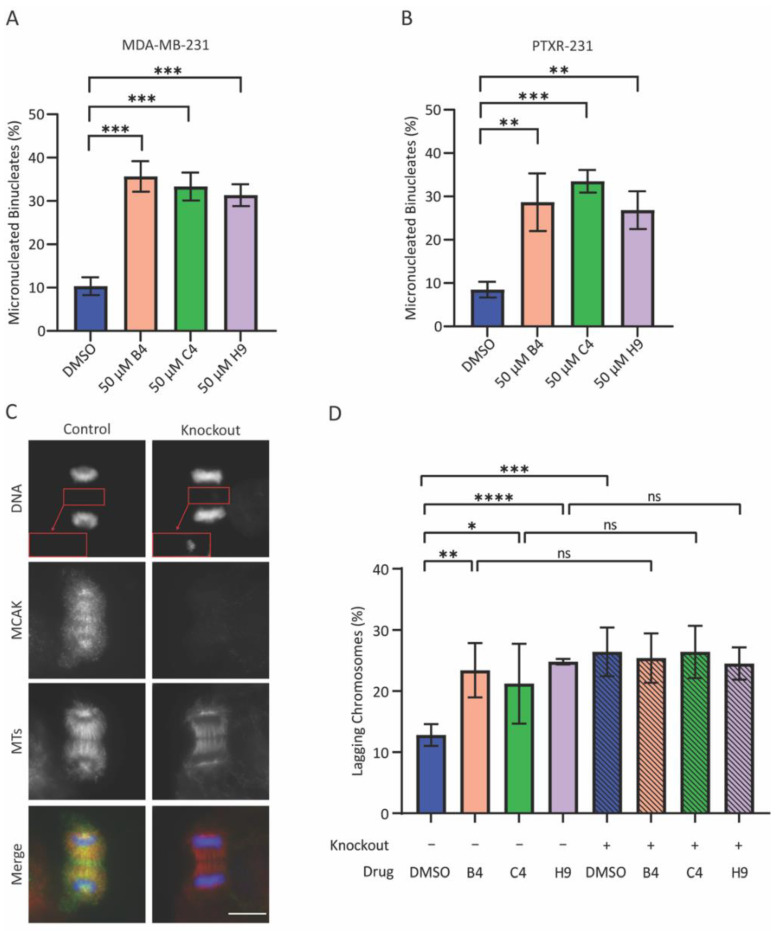
MCAK inhibitors induce aneuploidy in taxane-sensitive and -resistant cell lines. (**A**,**B**) Percentage of binucleate cells with micronuclei in MDA-MB-231 cells (**A**) or the taxane-resistant daughter line PTXR-231 (**B**). Values represent the mean +/− SD for each condition. Conditions were compared with a Student’s *t*-test. (**C**) Representative images of control (left) or MCAK knockout (right) cells in anaphase stained with antibodies to MCAK (green), MTs (red), and Hoechst 33342 to indicate DNA (blue). The red box (intensity adjusted three-fold higher) is an enlarged region between the segregating masses of DNA in which a lagging chromosome is present in the MCAK knockout cells (right). Scale bar = 10 µm. (**D**) Percentage of MCAK knockout HeLa cells in anaphase with lagging chromosomes in the absence (D, DMSO) and presence of the different compounds. Values represent the averages ± SD from three–four independent experiments. Conditions were compared with a Student’s *t*-test. * *p* < 0.05, ** *p* < 0.01, *** *p* < 0.001, **** *p* < 0.0001, ns, not significant.

**Figure 6 cancers-15-03309-f006:**
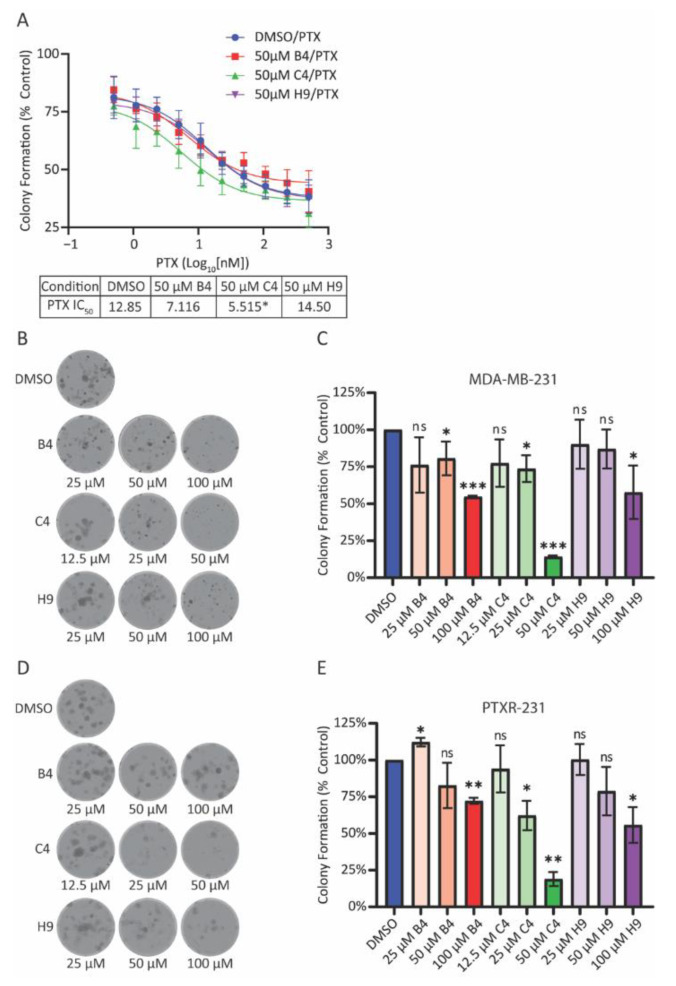
Inhibition of MCAK induces taxane sensitivity and reduces clonogenic growth. (**A**) Dose response curve and associated IC50 values for paclitaxel in MDA-MB-231 cells co-treated with 50 µM of either B4, C4, or H9 from five independent experiments. (**B**) Representative images of colonies of MDA-MB-231 cells treated with the specified concentration of inhibitor. (**C**) Colony formation percentage of MDA-MB-231 cells (n = 4). Percentages were compared to a value of 100% via a one sample *t*-test. (**D**) Representative images of colonies of PTXR-231 cells treated with the specified concentration of inhibitor as in (**B**). (**E**) Colony formation percentage in PTXR-231 (n = 3). Percentages were compared to a value of 100% via a one sample *t*-test. ns = not significant, * *p* < 0.05, ** *p* < 0.01, *** *p* < 0.001.

## Data Availability

No unique datasets were generated in this work. All primary data and analyses, and all materials generated are available upon request from the corresponding author.

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
