# Peer review of "MCAK Inhibitors Induce Aneuploidy in Triple-Negative Breast Cancer Models"

_cancers, 2023, doi:10.3390/cancers15133309_

Round 1

Reviewer 1 Report

The study performed by Smith et al. provides a valuable contribution to the field of potential targeted drugs for cancer treatment. The screen for chemical compounds that can be used as inhibitors for one of the members of Kinesin 13 family, namely Mitotic Centromere-Associated Kinesin (MCAK), is quite impressive. Though, I would rewrite the corresponding method section to make it more structured and clearer. Maybe moving methods details from results to methods can help.

I have few minor comments that authors should address in their revisions.

I do not understand why the authors used fluorescent version of MCAK designed for FRET experiments in their imaging-based MT depolymerization assay. Does FMCAK function similar to MCAK alone (without any tags at both termini)? I did not see any validation in the present manuscript. Can the authors address this issue?

In the experiments to verify that MCAK inhibitors induce aneuploidy in taxane-resistant cells the authors used inhibitors at the concentration of 50 uM. I wonder how they got this concentration if the original concentration of compounds was 25 uM (see line 424). Could the authors clarify?

I hope the authors specify somewhere what MCAK levels are in cell lines used in this study (HeLa, U2OS, MCF-10A, RPE-1, MDA-MB-231, PTXR-231).

Along with the schematic of MT depolymerization assay Fig 3A contains 2 images that have no description in the figure legend. What are they?

Fig 3B is not useful without additional information about structure of full-length MCAK, and the reference to its dimerization. One could only guess what Neck, CT, NT are.

I guess one should read ‘Compounds’ as ‘FMCAK + compounds’ in Figure 3 C, D, E. Otherwise, the plots do not match the experiments described in the manuscript.

Images in Fig 4A are hard too dark. I don’t know if that is the result of low resolution used for the first submission.

The enlarged lagging chromosome in Fig 5C looks suspiciously bright as compared to the boxed area without any zoom. Was it additionally manipulated?

Does ‘D’ in Fig 5D mean DMSO?

I think the paper could be useful for readers with different research background. They would appreciate if all abbreviations were spelled out at least once.

The authors use unconventional way to write concentrations: ‘mg mL-1’ (lines 389, 400, 437, 444, 460 …), ‘nm min-1’ (line 379), and equation (lines 411-412)

I am not sure what ‘0’ and ‘-1’ (log scale for paclitaxel concentrations) stand for in Figs. 2A-B and 6A, S2. And what is the base of these log scales?

Lines 65-66. In this content it is not obvious what ‘mitotic cells’ mean. The authors should be more explicative.

Figure legends should clearly state what panels show (plots, graphs, images) but not method descriptions and what analysis was performed.

Overall English is OK. 

Author Response

Reviewer 1

The study performed by Smith et al. provides a valuable contribution to the field of potential targeted drugs for cancer treatment. The screen for chemical compounds that can be used as inhibitors for one of the members of Kinesin 13 family, namely Mitotic Centromere-Associated Kinesin (MCAK), is quite impressive. Though, I would rewrite the corresponding method section to make it more structured and clearer. Maybe moving methods details from results to methods can help.

We thank the reviewer for their enthusiasm regarding our work. We have gone through the methods and checked for clarity. We paid strict attention to be sure that all of the experimental details were provided so that others can readily repeat our work. With regards to methods in the text, because part of this paper is about developing a screening assay, we felt that some of those details should be presented in the results to help the reader understand the logic of our assay design and screening. 

I have few minor comments that authors should address in their revisions. 

I do not understand why the authors used fluorescent version of MCAK designed for FRET experiments in their imaging-based MT depolymerization assay. Does FMCAK function similar to MCAK alone (without any tags at both termini)? I did not see any validation in the present manuscript. Can the authors address this issue?

Since a major premise of our screening design was to identify inhibitors that scored in two different assays for MCAK activity/function, we used the same exact protein in both screens to make them comparable. Previous work from our lab (Ems-McClung, et al. 2013) described the first MCAK conformational reporter and characterized its microtubule depolymerization activity, showing activity equivalent to previous studies. This work was the premise for the FRET screen presented in the current paper. We then made a conformational reporter for human MCAK with different fluorophores (GFP and mCherry; Zong et al., 2021) and used that to look at MCAK activity during cell migration. In other work from our lab (Zong, et al., 2016), we used the Ems-McClung et al. reporter in depletion/addback studies for spindle assembly reactions in Xenopus egg extracts and found that the fluorescently tagged MCAK fully rescues an MCAK depletion phenotype. Together, this prior work clearly establishes the functionality of these fluorescently tagged proteins.

We have updated the methods (lines 349-353) to clearly indicate that the fluorophores do not impact MCAK activity.

In the experiments to verify that MCAK inhibitors induce aneuploidy in taxane-resistant cells the authors used inhibitors at the concentration of 50 uM. I wonder how they got this concentration if the original concentration of compounds was 25 uM (see line 424). Could the authors clarify?

We apologize for the confusion. Drug stocks for the in vitro screening assays were supplied at a concentration of 25 µM in the assay plates. For the cell-based assays, we acquired the drugs in powder form from Chembridge and diluted them to stock concentrations of 50 mM or 100 mM for B4 or C4/H9 respectively to use in follow-up experiments.

The screening concentration of drug was used in vitro with purified proteins. When testing the compounds in cell-based assays, we tested several concentrations of drug and looked for effects on the induction of lagging chromosomes in anaphase. The 50 µM concentration caused a phenotypic response in line with what we see with MCAK knockdown. Given that we do not know the solubility, membrane permeability, or long-term stability in media, using a two-fold higher concentration did not seem unreasonable.

I hope the authors specify somewhere what MCAK levels are in cell lines used in this study (HeLa, U2OS, MCF-10A, RPE-1, MDA-MB-231, PTXR-231).

We assembled a new supplemental table (Table S1) that includes this data. Because there was not a single database that includes expression data for all of the cell lines, we assembled the data as relative expression compared to the MDA-MB-231 cells since every database included that cell line.  

Along with the schematic of MT depolymerization assay Fig 3A contains 2 images that have no description in the figure legend. What are they?

We apologize for this oversight, which is now clarified in the figure legend. The left image is a field of view from the imaged well, and the right image highlights what the FIJI algorithm detects.

Fig 3B is not useful without additional information about structure of full-length MCAK, and the reference to its dimerization. One could only guess what Neck, CT, NT are.

The figure legend has been edited to define what each domain represents. The functions of each domain have been well established through many prior studies, including the initial study describing the FRET reporter.

I guess one should read ‘Compounds’ as ‘FMCAK + compounds’ in Figure 3 C, D, E. Otherwise, the plots do not match the experiments described in the manuscript.

This is correct. Figure 3 has been revised to make this change.

Images in Fig 4A are hard too dark. I don’t know if that is the result of low resolution used for the first submission.

We adjusted the images such that the brightest pixels (which in some cases represent a small aggregate of tubulin) are not saturated. We have rescaled all images so that the microtubules are more clearly visible. A new version of Figure 4 has been included in the resubmission.

The enlarged lagging chromosome in Fig 5C looks suspiciously bright as compared to the boxed area without any zoom. Was it additionally manipulated?

We apologize, but this was a miscommunication between the PI and the primary author about scaling of the images and where/how to report the scaling. Both full size images are scaled identically for the DNA channel. This reviewer is indeed correct in that the boxed area in Figure 5 was both enlarged and presented with a different scaling to enhance contrast, which obviously begs the question of whether there are any dim chromosomes in the control condition. The figure legend should have clearly stated this. We went back to our original images to adjust the scaling of the primary image and then showed a boxed region in both the control and the MCAK inhibited cells that were enhanced equally.

Does ‘D’ in Fig 5D mean DMSO?

The reviewer is correct, and this has now been defined in the figure legend.

I think the paper could be useful for readers with different research background. They would appreciate if all abbreviations were spelled out at least once.

We have gone back over the paper to make sure that all abbreviations are defined upon first use.

The authors use unconventional way to write concentrations: ‘mg mL-1’ (lines 389, 400, 437, 444, 460 …), ‘nm min-1’ (line 379), and equation (lines 411-412)

We have changed all value to be mg/mL.

I am not sure what ‘0’ and ‘-1’ (log scale for paclitaxel concentrations) stand for in Figs. 2A-B and 6A, S2. And what is the base of these log scales?

The data is reported as log (base 10), which is the convention when people say “log” as opposed to natural log (ln). We have updated all axis labels to indicate log10 for the sake of clarity.

Lines 65-66. In this content it is not obvious what ‘mitotic cells’ mean. The authors should be more explicative.

We have changed the text to read cells in metaphase of mitosis.

Figure legends should clearly state what panels show (plots, graphs, images) but not method descriptions and what analysis was performed.

All figure legends have been edited to remove methodology.

Reviewer 2 Report

  1. ·         A brief summary: Smith et al., have penned a comprehensive report on the discovery and applications of novel cancer therapeutics, particularly targeting highly aggressive triple-negative breast cancer (TNBC), potentially increasing the lifespan of patients. In this article, the authors have identified three novel inhibitors of the Kinesin-13 MCAK (upregulated in TNBC), which in turn induce genome aneuploidy, an effect observed upon treatment with like cells treated with taxanes. This novel therapeutic strategy has been reported by the authors to sensitize the TNBC cells to taxanes with no neurodegenerative effects, thereby expanding the domain of precision medicine to improve patient outcomes in the future.

    ·         General concept comments:

    Review: The article is highly comprehensive with a clear rationale i.e., to develop drugs against targets that limit aneuploidy having minimal side effects and chemoresistance in TNBC patients. The experimental results are highly promising showing that MCAK C4 may serve as both a biomarker of prognosis and as a therapeutic target. The gap in current knowledge about RCC has been well identified and all the references have been appropriately cited.

    Specific comments: The following questions need to be answered to have a better understanding of this study and its potent future implications:

    1. Identifying the mechanism of action through structural (docking) and functional kinetics, i.e., whether they are catalytic inhibitors or allosteric inhibitors
    2. The control neuronal origin cell lines, such as neuron and glia cells, must be treated and tested to show minimal side effects of the potential MCAK inhibitors.
    3. Being an active cancer biologist and working on screening potential cancer therapeutics, I think the IC50 value of the research group’s best compound, C4, is 50 µM, which is in the higher range compared to the IC50 values of potential cancer therapeutics against TNBC and ER+ breast cancer.
    4. I suggest combinatorial therapies to determine the synergistic effects of novel MCAKi with radiation in breast cancer cell lines.
    5. Furthermore, in vivo studies with tumor xenograft models using either a mono- or dual-therapeutic approach need to be pursued in the future.
    6. The research article is well written in fluent English. My only suggestion is to remove the extra space in lines 274, 587, 722, and 757.

Author Response

A brief summary: Smith et al., have penned a comprehensive report on the discovery and applications of novel cancer therapeutics, particularly targeting highly aggressive triple-negative breast cancer (TNBC), potentially increasing the lifespan of patients. In this article, the authors have identified three novel inhibitors of the Kinesin-13 MCAK (upregulated in TNBC), which in turn induce genome aneuploidy, an effect observed upon treatment with like cells treated with taxanes. This novel therapeutic strategy has been reported by the authors to sensitize the TNBC cells to taxanes with no neurodegenerative effects, thereby expanding the domain of precision medicine to improve patient outcomes in the future.

      General concept comments:

Review: The article is highly comprehensive with a clear rationale i.e., to develop drugs against targets that limit aneuploidy having minimal side effects and chemoresistance in TNBC patients. The experimental results are highly promising showing that MCAK C4 may serve as both a biomarker of prognosis and as a therapeutic target. The gap in current knowledge about RCC has been well identified and all the references have been appropriately cited.

We thank this reviewer for valuing our contribution to the field.

Specific comments: The following questions need to be answered to have a better understanding of this study and its potent future implications:

1. Identifying the mechanism of action through structural (docking) and functional kinetics, i.e., whether they are catalytic inhibitors or allosteric inhibitors

We agree with this reviewer that understanding the mechanism of action of these drugs will ultimately be important. Our microtubule depolymerization assay measures the activity of the drugs by sedimenting samples onto a coverslip in a rotor that only holds six samples at a time (the 16 place rotor does not spin fast enough to sediment microtubules through a glycerol cushion), making this assay unsuitable for measuring the kinetics of microtubule depolymerization activity. More critically, solving the structure of the drug bound to the enzyme is the only true way to answer the structural question, something that is well beyond the scope of the current paper. Given that the structures of all three drugs are dissimilar, that none of the drugs resemble ATP (the substrate of the enzyme), and that all three drugs affect the conformation of MCAK as judged by the FRET assay, we favor the idea that they are allosteric inhibitors. We have revised the discussion to make this point clearer (lines 1174-1178).

2.  The control neuronal origin cell lines, such as neuron and glia cells, must be treated and tested to show minimal side effects of the potential MCAK inhibitors.

We assume that the reviewer is getting at the idea that MCAK inhibitors would not reduce peripheral neuropathy. It is important to point out that we currently do not know the mechanism by which taxanes (and other anti-microtubule agents) induce neuropathy, but it is likely complex and affects multiple types of peripheral nerves. Furthermore, most neuronal cells in the body are terminally differentiated, but neuronal cells in culture are proliferative, which would compound interpretation of any experiments done with inhibitors of a largely mitotic protein in cultured cells. It may well be that these inhibitors would induce aneuploidy in neuronal cells in culture without affecting neuropathy in the brain. These experiments will be much more informative when we have compounds that could be used in animal models of cancer and neuropathy. 

3. Being an active cancer biologist and working on screening potential cancer therapeutics, I think the IC50 value of the research group’s best compound, C4, is 50 µM, which is in the higher range compared to the IC50 values of potential cancer therapeutics against TNBC and ER+ breast cancer.

We agree completely with this reviewer on this point. Moving forward, a critical line of experimentation will be to screen chemical space to try and identify more potent analogs for drug development. We did analyze the Chembridge library for chemically related compounds and did not uncover any additional compounds to screen in our assays. Alternatively, we could use the assays we developed here to screen additional libraries for more potent inhibitors.  These limitations do not decrease the utility of the current compounds, which provide proof of principle to investigate how MCAK could be targeted therapeutically. 

4. I suggest combinatorial therapies to determine the synergistic effects of novel MCAKi with radiation in breast cancer cell lines.

Exploring how MCAK inhibition interacts with radiation is an appealing idea. The work by Zhu et al. (2020) provides evidence that MCAK may function in DNA repair, with laser irradiation as one of their main methods of inducing DNA damage. Understanding the mechanism of how MCAK contributes to DNA damage, whether the current compounds induce DNA damage and whether this is the mechanism by which MCAK synergizes with Taxanes, is the subject of the second half of the primary author’s thesis work. 

5. Furthermore, in vivo studies with tumor xenograft models using either a mono- or dual-therapeutic approach need to be pursued in the future.

We agree that this will be a valuable line of study once we have compound derivatives with a better pharmacological profile (more potent, known bioavailability).

6. The research article is well written in fluent English. My only suggestion is to remove the extra space in lines 274, 587, 722, and 757.

We will work with the journal staff to rectify these issues. We believe it was a file conversion error as they are not present in our original MS word file.